

# Excess soil moisture and fresh carbon input are prerequisites for methane production in podzolic soil

Mika Korkiakoski[1], Tiia Määttä[2], Krista Peltoniemi[2], Timo Penttilä[2], Annalea Lohila[1,3]

[1]Institute for Atmospheric and Earth System Research/Physics (INAR), Faculty of Science, P.O. Box 68, 00014 University of

Helsinki, Helsinki, Finland

[2]Natural Resources Institute Finland (Luke), Latokartanonkaari 9, 00790 Helsinki, Finland

[3]Finnish Meteorological Institute, P.O. Box 503, FI-00101 Helsinki, Finland

*Correspondence to*: Mika Korkiakoski (mika.korkiakoski@helsinki.fi)

**Abstract.**

Boreal upland forests are generally considered methane ($CH_4$) sinks due to the predominance of $CH_4$ oxidising bacteria over the methanogenic archaea. However, boreal upland forests can temporarily act as $CH_4$ sources during wet seasons or years. From a landscape perspective and in annual terms, this source can be significant as weather conditions may cause flooding, which can last a considerable proportion of the active season and because often, the forest coverage within a typical boreal catchment is much higher than that of wetlands. Processes and conditions which change mineral soils from acting as a weak

sink to a strong source are not well understood. We measured soil $CH_4$ fluxes from 20 different points from regularly irrigated and control plots during two growing seasons. We also estimated potential $CH_4$ production and oxidation rates in different soil layers and performed a laboratory experiment, where soil microcosms were subjected to different moisture levels and glucose addition simulating the fresh labile carbon (C) source from root exudates. The aim was to find the key controlling factors and conditions for boreal upland soil $CH_4$ production. Probably due to long dry periods in both summers, we did not find occasions

of $CH_4$ production following the excess irrigation, with one exception in July 2019 with emission of 18200 µg $CH_4$ $m^{-2}$ $h^{-1}$. Otherwise, the soil was always a $CH_4$ sink (median $CH_4$ uptake rate of 260–290 and 150–170 µg $CH_4$ $m^{-2}$ $h^{-1}$, in control and irrigated plots, respectively). The median soil $CH_4$ uptake rates at the irrigated plot were 88 % and 50 % lower than at the control plot in 2018 and 2019, respectively. Potential $CH_4$ production rates were highest in the organic layer (0.2–0.6 nmol $CH_4$ $g^{-1}$ $d^{-1}$), but some production was also observed in the leaching layer, whereas in other soil layers, the rates were negligible.

Potential $CH_4$ oxidation rates varied mainly within 10–40 nmol $CH_4$ $g^{-1}$ $d^{-1}$, except in deep soil and the organic layer in 2019, where potential oxidation rates were almost zero. The laboratory experiment revealed that high soil moisture alone does not turn upland forest soil into a $CH_4$ source. However, a simple C source, e.g. substrates coming from root exudates with high moisture switched the soil into a $CH_4$ source. Our unique study provides new insights into the processes and controlling factors on $CH_4$ production and oxidation and resulting net efflux, that should be incorporated in process models describing global $CH_4$

cycling.



## 1 Introduction

Methane (CH$_4$) is a greenhouse gas with a significant impact on the global climate. CH$_4$ increases the global temperatures by absorbing infrared radiation into its carbon-hydrogen bonds, resulting in a higher amount of heat energy within the atmosphere (e.g. Chai et al., 2016; Dlugokencky et al., 2011; Whalen, 2005). In soil, CH$_4$ is predominantly formed in biological anaerobic decomposition processes (Le Mer and Roger, 2001; Wuebbles and Hayhoe, 2002). Archaea called methanogens are responsible for the biological production of CH$_4$ in anoxic conditions, whereas methanotrophs conduct aerobic CH$_4$ oxidation (Hanson and Hanson, 1996; Orata et al., 2018; Thauer et al., 2008). The dynamics behind soil CH$_4$ sources and sinks depend on the ratio between CH$_4$ production and oxidation and its transport from the soil to the atmosphere, all of which are affected by an extensive network of numerous biotic and abiotic variables. The interannual fluctuations in global and regional CH$_4$ emissions are influenced by so-far largely unknown variables, the investigation of which is thus essential for understanding the changing dynamics in the current and future CH$_4$ budgets (Bousquet et al., 2006; Crill and Thornton, 2017; Dlugokencky et al., 2011; Fischer et al., 2008; Kirschke et al., 2013). The boreal zone in the northern hemisphere regularly presents large CH$_4$ emissions due to the abundance of anoxic wetlands, but part is counterbalanced by high oxidation rates in boreal upland forests. The CH$_4$ emission estimates from the boreal zone lie between 25 and 100 Tg yr−1, which combined with subarctic tundra environments account for approximately 3–10 % of the global CH$_4$ emissions (Olefeldt et al., 2013).

Boreal upland forests are broadly considered CH$_4$ sinks due to strongly oxic soils (Gulledge and Schimel, 2000; Megonigal and Guenther, 2008; Oertel et al., 2016; Whalen et al., 1991; Yavitt et al., 1990, 1995). In upland soils, high-affinity methanotrophs can consume CH$_4$ at atmospheric concentrations (Knief et al., 2003; Kolb, 2009). In addition, anaerobic CH$_4$ oxidation is known to occur in boreal forest soils (Blazewicz et al., 2012). Despite the abundance of oxygen in the boreal upland forest soil, there are some indications of smaller-scale CH$_4$-producing areas, such as wet depressions (Christiansen et al., 2012; Megonigal and Guenther, 2008; Vainio et al., 2021). In addition, some studies have found that upland forest soils may become CH$_4$ sources of varying significance after long periods of heavy precipitation (Lohila et al., 2016; Savage and Moore, 1997). Methanogenic population can stay constant in forest and other dry aerated soils and becomes active under wet and anoxic conditions (Angel et al., 2012; Peter Mayer and Conrad, 1990). With upland forests occupying a significant portion of the boreal zone, a more thorough examination of the complex dynamics behind the sink-source transitions of the forests is needed, especially in the context of climate change which may alter global and regional precipitation and temperature patterns (e.g. Beier et al., 2012; Lehtonen et al., 2014; Lohila et al., 2016). Lohila et al. (2016) also suggested that wet conditions can potentially affect the CH$_4$ exchange patterns differently in forests and wetlands by increasing and decreasing the CH$_4$ emissions in those ecosystems, respectively, amplifying the vital role of upland forests in the regional CH$_4$ balance in wet years. Furthermore, as precipitation may increase during summer and autumn in northern latitudes (Jylhä et al., 2009), this flooding-induced source of CH$_4$ may be activated more frequently in the future. This source is accounted for in the models of global CH$_4$ emissions, but there are recent observation-based indications that its magnitude may be severely underestimated, suggesting that the total annual emissions from upland forest soils in wet years may be nearly as large as those from northern



peatlands (Lohila et al., 2016; Saunois et al., 2020). It has already been suggested that the emissions from wet mineral soils can be the primary driver for the interannual variability in global $CH_4$ emissions (Spahni et al., 2011).

Soil temperature and moisture manipulations in $CH_4$ flux studies from upland soils have been very few, but some existing manipulation studies exist that focus on carbon dioxide ($CO_2$) fluxes (Allison and Treseder, 2008; Billings et al., 2000; Niinistö et al., 2004; Wu et al., 2011). Recommendations have been made to focus on precipitation manipulations carried out either by wetting or drying and establishing those experiments in mostly underrepresented forest ecosystems (Wu et al., 2011). Methanotrophs are known to be more sensitive to soil drying than methanogens (Ebrahimi and Or, 2018; Megonigal and Guenther, 2008). Since the processes and conditions that change mineral soils from $CH_4$ sink to a source are not sufficiently well understood, direct laboratory measurements of $CH_4$ formation in different soil layers under controlled temperature and moisture conditions are needed to explain the processes in mineral soil in greater detail.

In this study, changes in forest floor $CH_4$ fluxes were assessed with an irrigation experiment during the growing period in a boreal upland forest in Kenttärova in northern Finland. Kenttärova was chosen as the study site due to significant soil $CH_4$ emissions detected after a long period of abundant precipitation in 2011 by Lohila et al. (2016). In addition, $CH_4$ production and oxidation potentials were determined in different soil layers at flux measurement points. Finally, a laboratory microcosm experiment was used to investigate the conditions (temperature, moisture) needed to initialise $CH_4$ production from the upland soil. The aims of this study were 1) to find if the irrigation has any impact on the soil $CH_4$ flux and oxidation and production potentials; 2) to find which soil layers are most significant for $CH_4$ production and oxidation; and 3) to find the optimal conditions and key controlling factors for upland soil $CH_4$ production and oxidation. We hypothesised that 1) wet conditions prevailing for one or two summers could be seen in the response of microbial populations so that at the irrigated plot, the potential $CH_4$ oxidation would be smaller and at least short production episodes could be detected in the latter part of the summer either after both summers or at least after the second wet summer; 2) highest $CH_4$ oxidation potential are found in the surface soils while the maximum production potentials are found in the deeper layers; 3) both wet conditions and fresh organic carbon are needed to create conditions suitable for $CH_4$ production in podzolic forest soil.

## 2 Materials and methods

### 2.1 Study site

The study was carried out at the Kenttärova forest (67°59.237'N, 24°14.579'E) in the Kittilä municipality in Finland at the transition zone of the northern-boreal and subarctic zones (Fig. 1). The site is located on a hilltop plateau with an approximate elevation of 347 m above sea level and 60 m above the surrounding plains (Aurela et al., 2015). The study site has climatic and vegetational characteristics typical for a northern-boreal environment. The long-term (1981–2010) annual temperature and precipitation within the area are –1.0 °C and 521 mm, respectively, with long-term averages of January and July being –14 °C





and 14 °C (Pirinen et al., 2012). The maximum snow depth (average peak: 73 cm) is typically observed in late March; the median end date of snowmelt is May 14th and snow cover start date October 24th, respectively (Lohila et al., 2015). The soil type is podzol with glacial till as soil parent material (Aurela et al., 2015). Typical of the region and soil type, the site represents *Hylocomium-Myrtillus* type (HMT; Cajander, 1926; Ylläsjärvi and Kuuluvainen, 2009), *Picea abies* being the dominant tree

species mixed with a variety of some deciduous trees such as *Betula pubescens*, *Populus tremula*, and *Salix caprea*. The forest floor vegetation at Kenttärova consists primarily of forest shrubs, such as *Vaccinium myrtillus*, *Empetrum nigrum* and *Vaccinium vitis-idaea*, and a continuous and vigorous feather moss cover of *Pleurozium schreberi*, *Hylocomium splendens* and *Dicranum polysetum* with sporadic occurrences of lichens (Aurela et al., 2015). The dominant height of the uneven-aged (1–250 yr) tree stand reached approximately 15 m while the heights of individual spruce trees varied greatly. Some of the birches

at Kenttärova were logged for firewood in the 1960s but since then the forest has grown without human disturbances (Aurela et al., 2015).

## 2.2 Experimental setup

For examining causal relationships between $CH_4$ flux and soil moisture and temperature, the field study included two plots: irrigation ($S_i$) and control ($S_c$) without irrigation treatment (Fig. 1). The surface areas of $S_c$ and $S_i$ were approximately 280 m$^2$

and 120 m$^2$, respectively. Both plots included ten measurement points. Measurement points were assigned somewhat randomly in both plots, with the aim to represent as similar vegetational, topographical and sun aspect characteristics as possible. Both the $S_c$ and $S_i$ and measurement points were connected with wooden boardwalks to minimise soil and vegetation disturbance from trampling.

Soil moisture was manipulated by irrigating part of the experimental area with two water sprinklers. The irrigation periods

were 28 May 2018–7 September 2018 and 6 June 2019–29 August 2019. The sprinklers were set in the plot so that the irrigated water would evenly reach each measurement point. The irrigated area in practice reached approximately 118 m$^2$ with 3–5.5 m width and 10–21 m length, depending on the wind conditions. The amount of irrigated water was 2 x 1000 l a week during 28 May–1 June 2018, after which the amount was increased to 3 x 1000 l a week during 7–18 June 2018 and eventually to 5 x 2000 l a week during 20 June 2018–7 September 2018. In 2019, the plot was irrigated 3 x 1000 l a week throughout the

summer. For ensuring a relatively even distribution of irrigated water in the plot, the spatial distribution of irrigation was checked with rain gauges (unit: mm) and plastic buckets. The amount of water in each bucket was later proportioned to the rain gauges in mm based on their dimensions. The precipitated water was measured after each irrigation from the end of May 2018 to mid-June 2018, after which the precipitated water was measured only when the weather was notably windy and/or natural rainfall occurred during the irrigation. It was estimated that 1000 l irrigation resulted on average to 11 mm and 2000 l

to 21 mm of precipitation.





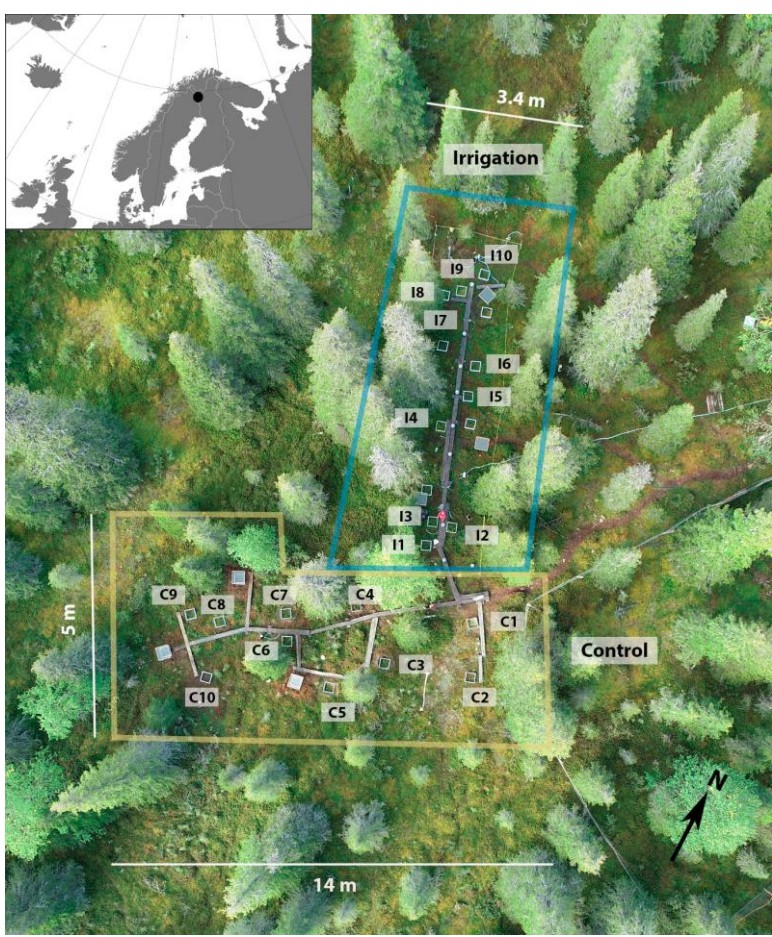

Figure 1. Experimental setup and location of the study site. Aerial image by Dr. Bastian Steinhoff-Knopp (Leibniz University Hannover, September 2018).

## 2.3 CH$_4$ flux measurements and calculation

5  Chamber measurements started on 30 May 2017 on eight measurement points, of which four were located on S$_i$ and S$_c$, respectively. Six additional points were added to both S$_i$ and S$_c$, and the measurements from these points started on 29 May 2018. The measurements were made mainly between June-September every two weeks in 2017 and weekly in 2018 and 2019. The measurements ended on 19 September 2019.

CH$_4$ fluxes were measured with 5 min closure time on the forest floor by the closed-chamber system with an opaque rectangular

10  chamber (60 x 60 x 20 cm, length x width x height). The chamber included a fan to mix the air inside the chamber and a vent tube to prevent pressure differences between the chamber headspace and the atmosphere. Also, chamber headspace temperature was recorded with HOBO Pendant Temperature Data Logger (Onset Computer Corporation, Bourne, MA, USA).





The bottom of the chamber edges had a foam layer to prevent leakage between the collar and the chamber. All the measurement points had metal collars (58 x 58 x 30 cm, length x width x height) installed about 2 cm deep into the soil. $CH_4$ and water vapour ($H_2O$) mixing ratios were measured with G2301 (Picarro Inc., CA, USA) and G1301-m (Picarro Inc., CA, USA) before and after 28 June 2018, respectively. The gas analyser was located inside a cabin about 20 m away from the measurement

point. The gas sample from inside the chamber was transported to the analyser by 20 m long tubing (inner diameter 3.1 mm, Bevaline IV) with a 1 l min$^{-1}$ flow rate where the mixing ratio was sampled every 3–4 seconds. The sampled gas was not returned to the chamber, which causes underpressure inside the chamber and underestimating the flux estimation. Because the chambers had a vent tube, we corrected the leakage with an assumption that the underpressure consisted of ambient air.

$CH_4$ fluxes were calculated as:

$$F = \left(\frac{dC(t)}{dt}\right)_{t=0} \frac{MPV}{RTA}, \quad (1)$$

where $\left(\frac{dC(t)}{dt}\right)_{t=0}$ is the concentration change over time from an exponential model (e.g. Korkiakoski et al., 2017) at the beginning of the closure, $M$ is the molecular mass of $CH_4$ or $N_2O$ (16.04 and 44.01 g mol$^{-1}$, respectively), $P$ is air pressure, $R$ is the universal gas constant (8.314 J mol$^{-1}$ K$^{-1}$), $T$ is the mean chamber headspace temperature during the closure, and $V$ is the air volume of the chamber and the collar, and $A$ is the base area of the chamber or collar. The snow depth and the height

of mosses and other vegetation in the chamber headspace volume were taken into account, ignoring the pore space in the soil and snow. The height of the vegetation was measured once a summer. The vegetation height was assumed to remain constant for that year.

When calculating the $CH_4$ balances, measured $CH_4$ fluxes were assumed to be daily mean fluxes. The gaps in the data were filled by linear interpolation. To avoid a biased 2019 balance estimate for point I1, the $CH_4$ emission peak observed on 27 June

2019 was ignored when calculating the balance.

The micrometeorological sign convention is used throughout the paper: a positive flux indicates a flux from the ecosystem to the atmosphere (net emission), and a negative flux indicates a flux from the atmosphere into the ecosystem (net uptake).

### 2.4 $CH_4$ production and oxidation potential measurements

Samples for the potential $CH_4$ production and oxidation were taken on 23 August 2018 and 26 August 2019. Six composite

samples were collected from both $S_i$ and $S_c$ next to the chamber collars. Composite soil samples were combined from 3-5 core samples taken by soil auger separating four soil horizons: the organic layer without vegetation (O) and the three mineral soil layers below (zone of eluviation, i.e., leaching layer, E; zone of illuviation, i.e., enrichment layer, I; C-horizon representing the bottom layer, C). Samples were kept at 4 °C during the shipment into the lab and before analyses. The mean depths of the



soil layers were 5.7 cm, 10.5 cm, 18.9 cm, and 32.3 cm, while the mean thicknesses were 5.7 cm, 4.8 cm, 8.4 cm, and 13.4 cm for O, E, I and C layers, respectively. The layer depths and thicknesses were determined from 6 spots inside the experimental area.

Soil moisture and organic matter contents of the samples were determined with TGA-analyser (LECO TGA-701, Leco Corporation, USA) with the standard method (ISO11465), which measures weight loss as a function of temperature in a controlled environment. Soil pH was determined from methane oxidation bottles after measurement by increasing the ratio of 1:3 of deionised $H_2O$ and measuring them after 24 hours. Average soil pH, soil moisture and organic matter contents for the 2018 and 2019 samples are presented in Table S1. Total nutrients and C and N contents were determined from soil samples taken in 2018 with standard methods (ISO11466, 10694, 13878). Samples for the total nutrients were digested by the closed wet HNO3-HCl digestion method in a microwave (CEM MDS 2000), and the extract was analysed by iCAP 6500 DUO ICP-emission spectrometer (Thermo Scientific, UK). Total C and N were measured from sieved and air-dried samples on a CN analyser (Leco-TruMac, Leco Corp., St. Joseph, MI). Total nutrient, C and N contents for the year 2018 samples are shown in Table S2.

Fresh sieved soil (with 2 mm mesh size) was placed into 120 ml sterile incubation bottles with a standardised volume-based measuring scoop (20 ml). 10 ppm of $CH_4$ were added as a substrate into the bottles for determining potential $CH_4$ oxidation rates. Oxidation was measured by gas chromatograph (GC) for 24 hours. Two volumes of deionised $H_2O$ were added into production potential bottles and incubated two times with pure $N_2$ gas to remove oxygen and create anoxic conditions. Production bottles were measured by GC first twice and then once a week for 42 days to detect productions. Potential rates were calculated from the linear part of the curve showing the decrease or increase of $CH_4$ concentrations in time. The final potential rates are presented as nmol $CH_4$ $g^{-1}$ (dry mass of soil) $d^{-1}$.

**2.5 Microcosm experiment**

A microcosm experiment was designed to determine the conditions (temperature and moisture) that are needed to initialise the $CH_4$ production from the soil. For the experiment, soil profile samples were taken from the pit next to the $S_i$. Artificial soil profiles were constructed into the plastic jars (volume of 1.6 l), including the vegetation and organic layer and two mineral soils layers (leaching and enrichment layers). Half of the jar volume was left empty for headspace measurements. Jars were placed into two different growth chambers (Binder KBW, Germany) with two different temperatures at 15 °C and 25 °C. Both temperature conditions had three replicate jars, including controls without moisture increase (C) and two different levels of moisture increase, lower (M1) and higher (M2) moisture. The experiment also included separate triplicate jars with glucose added into controls (Cglu) and moisture increase (M1glu, M2glu) treatments. Glucose was added at the beginning of the measurements to simulate the effect of fresh, simple C source for microbes such as existing in root exudates. Added glucose amounts were adjusted to contain two times more C that is approximated to be bound into microbial biomass in forest soils to





see the possible effect. Briefly, it is generally assumed that about 3 % of the total C in forest soils is of microbial origin. Based on the total C content determined from 2018 samples, we calculated that the soil in jars would contain about 1 % of microbial C. Two moisture conditions for the jars were adjusted to be different enough to detect changes between the treatments. Average final moisture conditions were adjusted so that in the jars, the lower moisture content (M1) was about 50 % and the higher

content (M2) 80 % and the control jars (C) represented the average moisture content in the soil, which was about 30–35 % (Table 2).

Light conditions in the growth chambers were adjusted to mimic the natural light conditions at the end of August in northern Finland (about 15 hours light and 9 hours dark). Every week, the jars were switched from one growth chamber to another to avoid the differences due to features in the chamber itself. Moisture conditions were kept constant by weighing the jars twice

a week and adding the water to minimise the effect of evaporation. $CH_4$ fluxes were measured from the headspace of the jars once a week with LI-7810 (LI-COR Biosciences, NE, USA). The fluxes for the five-week measurement period were calculated from the exponential model the same way described in Chapter 2.3.

## 2.6 Soil temperature and moisture measurements

Multiple soil temperature (ST) and moisture (SM) sensors were used to record said variables next to the $CH_4$ flux measurement

points. ST was measured with 10 HOBO Pendant data loggers (Onset Computer Corporation, MA, USA) and SM with 9 EC5 Soil Moisture Smart Sensor (Onset Computer Corporation, MA, USA) with HOBO U30 USB Weather Station Data Logger (Onset Computer Corporation, MA, USA). In addition, 7 Soil Scout online sensors (Soil Scout Ltd, Helsinki, Finland) were used to measure both ST and SM. The time intervals for ST logging were 20 and 30 minutes for Soil Scouts and HOBO sensors, respectively. All the sensors were installed during 23 May – 6 June 2018 5 cm below the soil surface in the mineral

soil layer next to the collar and covered carefully with soil. The measurements continued until the experiment ended, except the SM measurements made with EC5 sensors, which broke down at the beginning of June 2019. The locations of the installed sensors are listed in Table S3.

SM was also measured from two different locations about 10 m distance from the $S_c$ and $S_i$. In both locations, SM was measured at 5 and 20 cm depths with ThetaProbe soil moisture sensor (Type ML2, Delta-T Devices Ltd, Cambridge, UK). In addition,

ST was measured next to one of the soil moisture sensors at 5 cm depth (PT100, PT4T, Nokeval Oy, Nokia, Finland).

## 2.7 Statistical methods

Fluxes between the different moisture levels and glucose addition in the microcosm experiment were compared by using the one-way analysis of variance (ANOVA) by using aov command in R programming language (R Core Team, 2021, v4.0.5). The same method was used for comparing the $CH_4$ production and oxidation potentials between the plots and years. In the

microcosm experiment, the glucose addition was compared only to the sample without added glucose on the same moisture





level and temperature. The effect of three different moisture levels was compared separately for added glucose and without added glucose groups by using Tukey's honestly significant difference (HSD) by 'multcomp' package in R (v1.4-14; Hothorn et al., 2008).

Linear mixed-effect model with Tukey's HSD post hoc test was used for testing the statistical significance of differences in
$CH_4$ fluxes between the $S_i$ and $S_c$. The linear mixed-effect model was carried out with the R programming language using 'lme4' package (Bates et al., 2015). The chamber points were treated as a random effect. The normality of the model residuals was visually checked using the quantile-quantile plot (Q-Q plot) method.

The linear mixed-effect model was also used for finding the most significant variables affecting $CH_4$ fluxes ($F_{CH4}$). The variables used in the modelling were: 5 cm soil temperature (ST) and moisture (SM), $CH_4$ oxidation potential ($OP_{CH4, x}$, where
x is one of O, E, I soil layers or the mean of all layers), $CH_4$ production potential ($PP_{CH4, x}$, where x is one of O, E, I soil layers or the mean), and carbon and nitrogen content ($CC_x$ or $NC_x$ where x is one of O, E, I soil layers or the mean). The model runs were divided into four parts: using mean values of all soil layers, using only values of a specific soil layer, combining values of multiple different soil layers. Even though SM and temperature were only measured at 5 cm depth, they were included in all the model runs. Measurement points were always treated as a random effect (u). The best model was selected by using
stepwise selection. We started with a full model and reduced the number of variables one by one using the Akaike information criterion (AIC) as the criteria, which was conducted using the drop1 function in R. The initial model in all but the combination model run was:

$$F_{CH4} = \beta_0 + \beta_1 ST + \beta_2 SM + \beta_3 PP_{CH4,x} + \beta_4 OP_{CH4,x} + \beta_5 CC_x + \beta_6 NC_x + \beta_7(u + e)$$

where e is the model error, $\beta_0$ is the model's intercept, and parameters from $\beta_1$ to $\beta_7$ are the regression coefficients of the
explaining variables. We used a 95 % confidence interval ($p < 0.05$) to determine whether the results were statistically significant.

Pearson correlation matrix including potential $CH_4$ production and oxidation rates and soil data (SM, organic matter, pH, nutrient elements) were created using commands rcor and corrplot in R. Significance level for correlation coefficients between variables was $p = 0.01$. In addition, simple linear regressions at 95 % confidence level with Pearson's correlation coefficient
and smoothed marginal histograms were used for primary correlation analyses between $CH_4$ flux and SM and $CH_4$ flux and ST using 'ggpubr' (v0.4.0; Kassambara, 2020) and 'cowplot' (v1.1.1; Wilke, 2020) packages in R.

### 2.8 Meteorological conditions

The mean air temperatures in the May–September period were 8.0, 11.0, and 8.9 °C for 2017-2019. Compared to the long-term (1981–2010; Pirinen et al., 2012) mean temperature of the same period (9.3 °C), 2017 was cooler and 2018 warmer than the
average, respectively. In 2019, the monthly temperatures during the measurement period were close to long-term averages



(Table S4). 2017 was the coolest year of the measurement period, primarily attributed to much cooler May and slightly cooler August than other years (Table S4). On the other hand, 2018 was the warmest year, primarily due to much warmer May and July. July 2018 was exceptionally warm (18.8 °C) compared to other years (2017: 13.0 °C, 2019: 12.6 °C) and long-term mean (13.9 °C).

5   The precipitation sums in the May-September period were higher than the long-term average (296 mm) in 2017 (335 mm) and 2019 (357 mm), but about the same in 2018 (293 mm). However, there were notable differences when inspecting monthly precipitation sums. In 2017, May, June and September were drier than in 2018 and 2019 (Table S4). On the other hand, in July 2017, the amount of precipitation (129 mm) was about 100 mm higher than in 2018 (28 mm) and 2019 (33 mm). Therefore, 2017 was markedly wetter compared to the long-term average in July (75 mm). On the other hand, 2018 and 2019 were

10  markedly drier than on average. In 2019 excluding July, the monthly precipitation sums were very similar and higher than the long-term mean.

The snow cover melted on 9 June 2017, 21 May 2018, and 26 May 2019. In 2017 and 2019, the first measurement day was made when snow was still on the ground (Fig. 2).

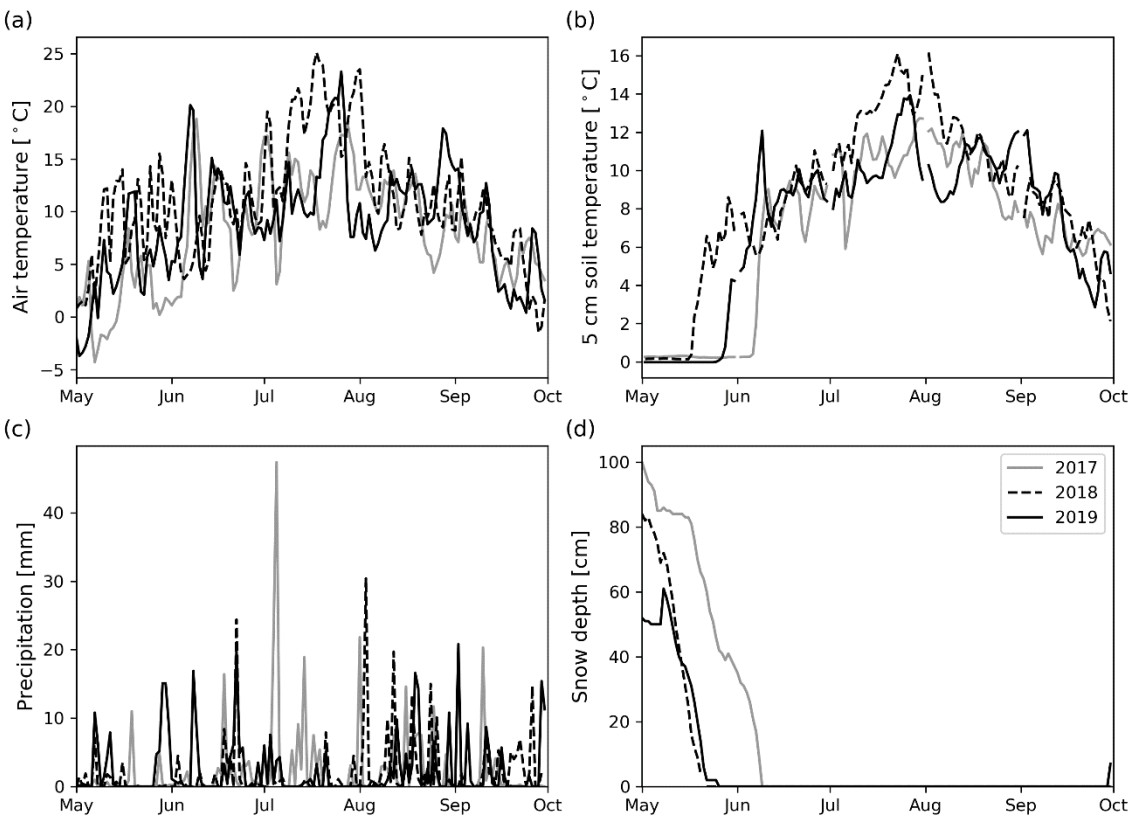





Figure 2. Daily mean air temperature (a), daily mean 5 cm soil temperature (b), daily precipitation sum (c) and daily snow depth (d) measured at Kenttärova weather station in May-September 2017-2019.

## 3 Results

### 3.1 Impact of irrigation on soil moisture and temperature

The growing seasons of the study years (2018 and 2019) were generally dry, based on the soil moisture data collected in long-term pits near the experimental area (Fig. 3). While in 2019, the whole growing season was dry, in 2018, the driest month was July. On the other hand, August was relatively wet in terms of precipitation (Table S4), but after a severe drought, the high precipitation was not enough to increase the soil moisture to the same range observed in 2017. There were large differences in SM profiles (located outside the experimental area) between the years (Fig. 3). In 2017, 5 cm soil moisture ($SM_{5cm}$) mainly

remained between 25–35 v%. Also, $SM_{5cm}$ in 2019 was relatively stable, varying within 15–20 v%, but it was markedly lower than in the other years, except in July 2018. $SM_{5cm}$ in 2018 had much temporal variation. In May 2018, $SM_{5cm}$ rose to 50 v% but fell quickly to 25 % after the snow had melted. In July 2018, the $SM_{5cm}$ fell quickly below 15 v% and kept decreasing down to 12 % until the beginning of May 2019, after which it started recovering up to 25 % until the measurement period ended in September 2019. In terms of absolute values, 20 cm soil moisture ($SM_{20cm}$) did not differ between years compared to

$SM_{5cm}$. In June, August and September, the $SM_{20cm}$ did not usually differ more than 3 v% between the years. In May, the rapid increases and decreases in $SM_{20cm}$ associated with snowmelt occurred at different strengths and times. In July, $SM_{20cm}$ in 2017 was about 5 % higher than in the other years, but the first half of August 2019 was drier than the other years.

At the experimental area, $SM_{5cm}$ was on average 6.5 v% lower at the $S_c$ than at the $S_i$ in June-September 2018. $SM_{5cm}$ in 2018 varied typically within 14–23 v% and 6–14 v% at the $S_i$ and $S_c$, respectively. However, one SM sensor measured about 10 %

higher values than the other sensors at the $S_c$ (Fig. 4). Also, at the beginning of August, $SM_{5cm}$ increased at one of the measurement points by 5 %. $SM_{5cm}$ remained on that higher level until the end of the measurement period in 2018. In 2019, $SM_{5cm}$ at the $S_i$ remained within 15–18 %, except in August and September.

In 2018, irrigation was performed on weekdays and during irrigation $SM_{5cm}$ rose by 10–15 v% (Fig. 4). However, the $SM_{5cm}$ decreased fast and usually returned to the pre-irrigation level before the next irrigation 24 hours later. In 2019, the rise of 5 cm

$SM_{5cm}$ due to irrigation was usually between 2–5 %.

5 cm daily mean soil temperatures ($ST_{5cm}$) were on average 0.7 °C higher at the $S_i$ compared to the $S_c$ in June-September 2018 (Fig. 5). Also, spatial variation was higher at the $S_c$ (Fig.). The biggest difference in daily mean $ST_{5cm}$ between the plots was observed around mid-July 2018 when the $ST_{5cm}$ at the $S_i$ was on average 2.0 °C higher than at the $S_c$. However, in 2019, the difference in daily mean $ST_{5cm}$ between the plots was small, and the $S_i$ was only about 0.2 °C warmer on average than the $S_c$.

30 Also, the maximum difference between the plots was about 0.6 °C, which occurred at the end of July and August.





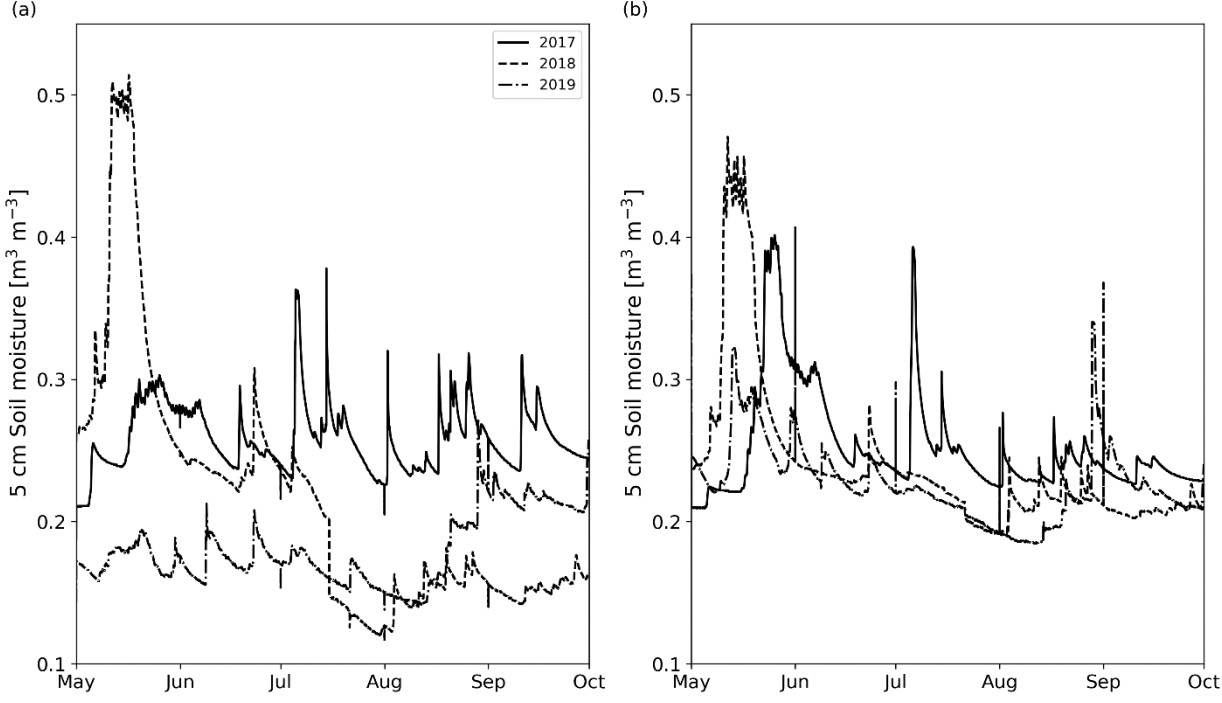

Figure 3. The daily mean 5 (a) and 20 (b) cm soil moisture time series measured from two different locations outside the experimental area from May to October in 2017 (solid line), 2018 (dashed line) and 2019 (dot-dashed).



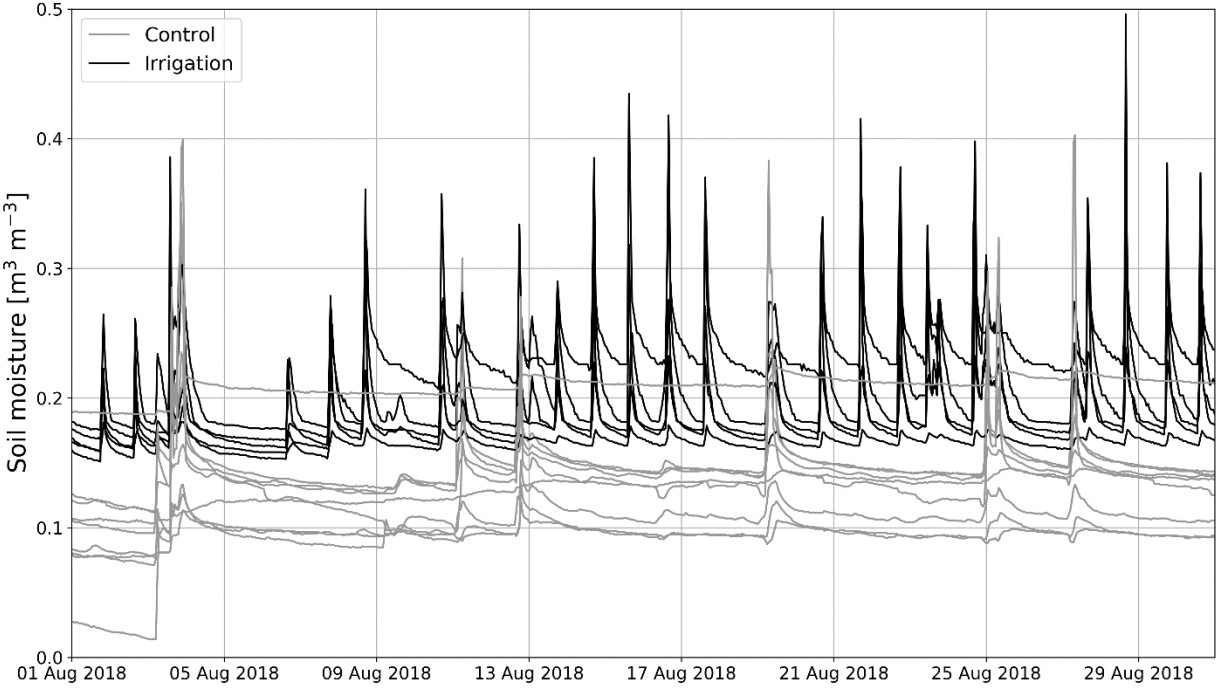

Figure 4. Hourly mean soil moisture time series measured at control (grey) and irrigation (black) plots in August 2018.

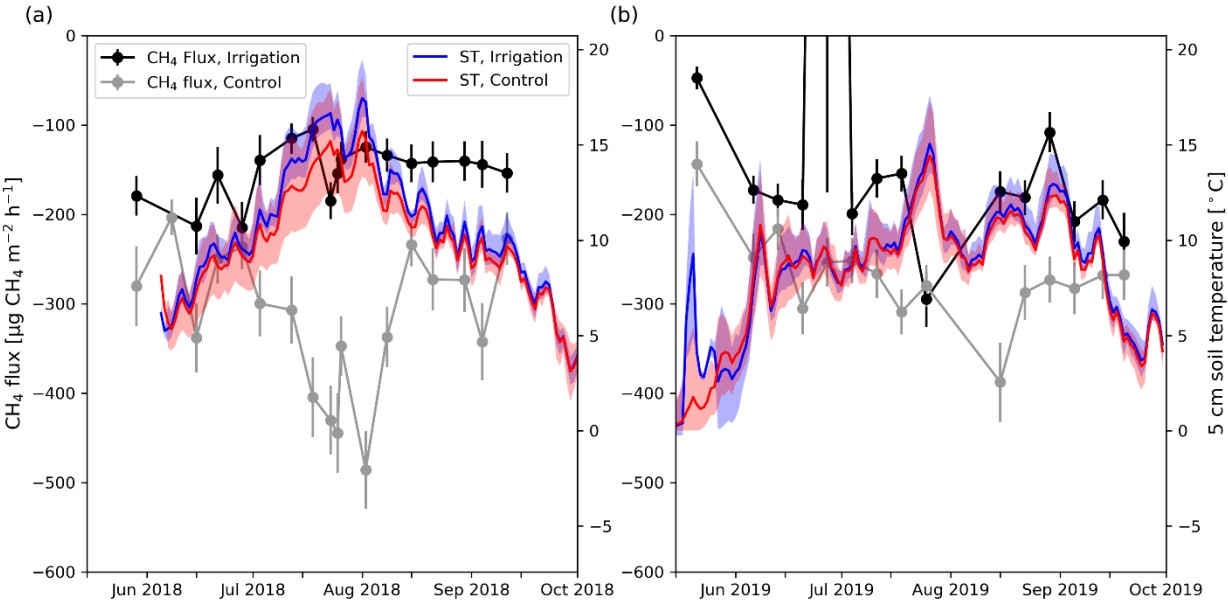





Figure 5. Daily mean $CH_4$ flux measured at the irrigated (black) and control (grey) plots in 2018 (a) and 2019 (b). The error bars show the standard error of the mean. Blue (irrigated) and red (control) lines represent daily mean 5 cm soil temperature (ST) and shading shows the minimum and maximum daily values measured by different sensors (irrigation: n=7, control: n=9).

### 3.2 The effect of irrigation on $CH_4$ uptake

Before the irrigation experiment started, all the measured $CH_4$ fluxes were negative, indicating $CH_4$ uptake, and did not differ significantly between $S_c$ and $S_i$. In 2017, the fluxes were measured from four points at each plot and median fluxes ($S_c$: –220 µg $CH_4$ $m^{-2}$ $h^{-1}$, $S_i$: –230 µg $CH_4$ $m^{-2}$ $h^{-1}$) and mean June-September $CH_4$ balances were similar between the plots (Fig. 6). However, there was notable spatial variation between the points as the June-September $CH_4$ balances varied between –950 and –470 mg $CH_4$ $m^{-2}$).

In 2018 and 2019, when irrigation started, the fluxes measured at the $S_i$ and $S_c$ differed significantly from each other in terms of long-term points (I2, I3, I4, I9, C1, C4, C6, C8; 2018: $p < 0.001$, 2019: $p = 0.01$). The mean summertime $CH_4$ uptake rates of long-term points in 2018 were 37 % larger and 15 % smaller than in 2017 at $S_c$ and $S_i$, respectively (Fig. 6; Table S5). In 2019, the mean June-September balances ($S_c$: –940±120 mg $CH_4$ $m^{-2}$, $S_i$: –660±70 mg $CH_4$ $m^{-2}$; Fig. 6; Table S5) remained at about the same level as in 2018 and the fluxes did not differ significantly from fluxes measured in 2018.

The median measured flux (Fig. 6; Table S5) across all the measurement points at the $S_i$ (–150 µg $CH_4$ $m^{-2}$ $h^{-1}$) was 88 % higher than at the $S_c$ (–290 µg $CH_4$ $m^{-2}$ $h^{-1}$) in 2018 and 50 % higher in 2019 ($S_i$: –170 µg $CH_4$ $m^{-2}$ $h^{-1}$, $S_c$: –260 µg $CH_4$ $m^{-2}$ $h^{-1}$). The fluxes differed significantly between the plots in both years (p < 0.001). All but one of the measured fluxes were negative, indicating $CH_4$ uptake. One large $CH_4$ emission case (18200 µg $CH_4$ $m^{-2}$ $h^{-1}$) was observed in point I1 on June 27[th] 2019. Similar differences were also observed in mean four-month (June – September) $CH_4$ balances (Table S5). There was 20  lots of spatial variation between the measurement points. $CH_4$ balances varied from –1280 mg $CH_4$ $m^{-2}$ to –480 mg $CH_4$ $m^{-2}$ at the $S_c$ and from –740 mg $CH_4$ $m^{-2}$ to –180 mg $CH_4$ $m^{-2}$ at the $S_i$ in 2018. Some of the measurement points at the $S_i$ had higher $CH_4$ uptake rates than some points located at the $S_c$, but on average $CH_4$ uptake rate was noticeably larger at the $S_c$ (–850±80 mg $CH_4$ $m^{-2}$) than at the $S_i$ (–450±60 mg $CH_4$ $m^{-2}$). In 2019, $CH_4$ uptake rates increased in most of the points at the $S_i$, averaging at –570±60 mg $CH_4$ $m^{-2}$, but the balances remained mostly the same at the $S_c$ (mean: –830±70 mg $CH_4$ $m^{-2}$).



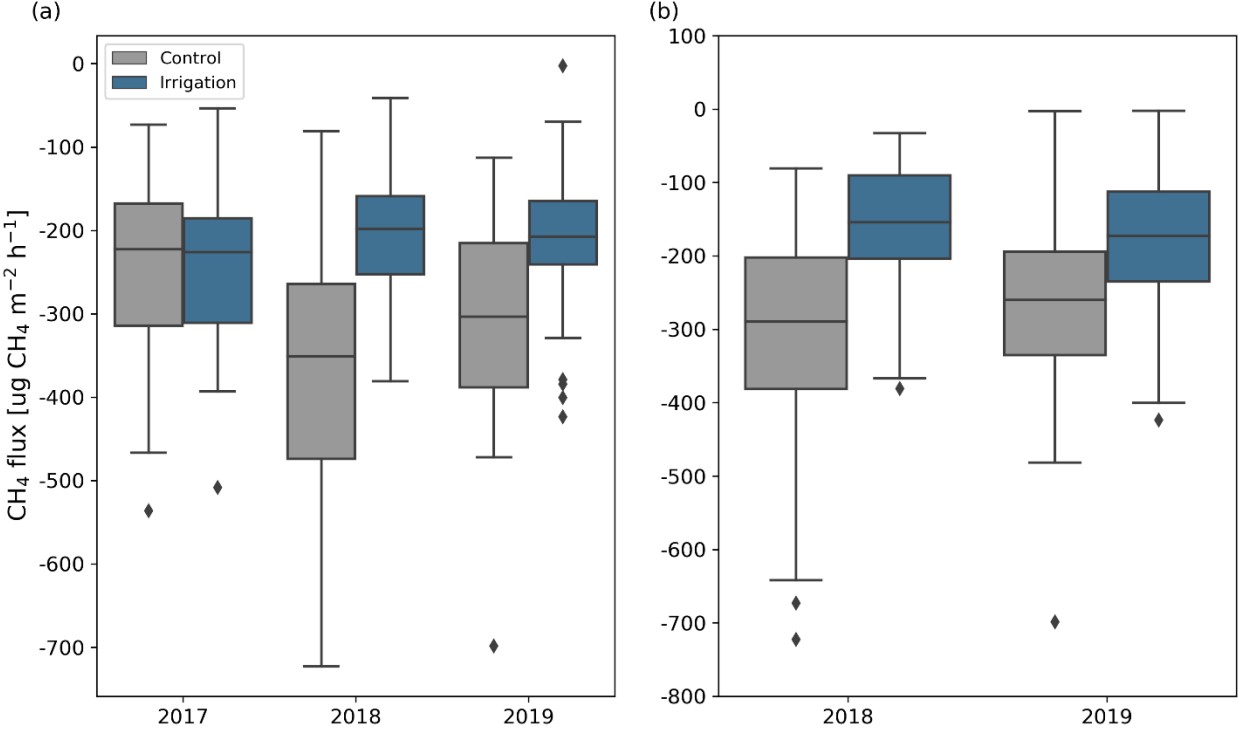

Figure 6. CH$_4$ fluxes measured by long-term (C1, C4, C6, C8, I2, I3, I4, I9) (a) and all (b) chamber points in May-September in different years. For comparison, the flux of the points located on the irrigated plot in 2018 and 2019 have been calculated already for 2017, even though the irrigation setup was established only in 2018. The boxes show the quartiles of the dataset and the horizontal line inside the boxes is the median flux. Whiskers show the range of the data, except for the points that are determined to be outliers, which are shown with black diamonds.

### 3.3 CH$_4$ production and oxidation potentials

Oxidation potential rates were quite similar in all soil layers, except for C-layer where the rates were lower. Between the years, however, the rates differed as those in 2019 were generally higher and more variable than in 2018 (Fig. 7ab). The most notable increase was detected in the organic layer, where the oxidation potential rates were mainly non-existent in 2018, but about 15 nmol CH$_4$ g$^{-1}$ d$^{-1}$ in 2019 at both S$_i$ and S$_c$. However, the change was significant only at the S$_i$ ($p = 0.03$). Oxidation rates were significantly ($p = 0.03$) higher in 2019 (median: 22 nmol CH$_4$ g$^{-1}$ d$^{-1}$) than in 2018 (median: 15 nmol CH$_4$ g$^{-1}$ d$^{-1}$) also in I layer at the S$_c$, but there was no significant difference in the same layer at the S$_i$. There were no statistically significant differences between the years in any other soil layers at either plot. Comparing the soil layers between the plots revealed that the oxidation rates were significantly higher ($p = 0.01$) in the C layer at the S$_c$ than in S$_i$ in 2018. The rates were significantly higher ($p < 0.01$) at the S$_c$ in I layer in 2019, but there were no other significant differences between the plots in other soil layers.



The highest $CH_4$ production potential rates occurred in O layer and some production potential was observed in E layer, while in the lowest soil layers, the production rates were negligible (Fig. 7cd). At the $S_i$, the production potential rates were significantly lower in 2019 than in 2018 in O ($p = 0.04$) and I ($p = 0.001$) layers. At the $S_c$, the production rates differed significantly ($p < 0.02$) only in the C layer, but the rates were negligible in both years. Comparing the production rates between the plots revealed that the rates were significantly ($p < 0.04$) higher in O layer at the $S_c$ in 2019. A significant difference ($p < 0.01$) between the plots was also found in the C layer in 2018.

Potential $CH_4$ production rates in 2018 had strong positive Pearson correlation coefficients ($\rho$) with organic matter content ($\rho = 0.94$), moisture contents ($\rho = 0.90$), total N and C amounts ($\rho = 0.95$ and $0.94$, respectively) and with several nutrient elements such as with sulphur ($\rho = 0.93$), lead ($\rho = 0.92$) and potassium concentrations ($\rho = 0.81$) determined from the soil samples (Fig. S1a). Potential $CH_4$ production rates in 2019 had a similar stronger positive correlation with organic matter ($\rho = 0.96$) and moisture contents ($\rho = 0.93$; Fig. S1b). Potential $CH_4$ oxidation rates showed only a negative correlation with measured copper concentrations ($\rho = -0.61$).

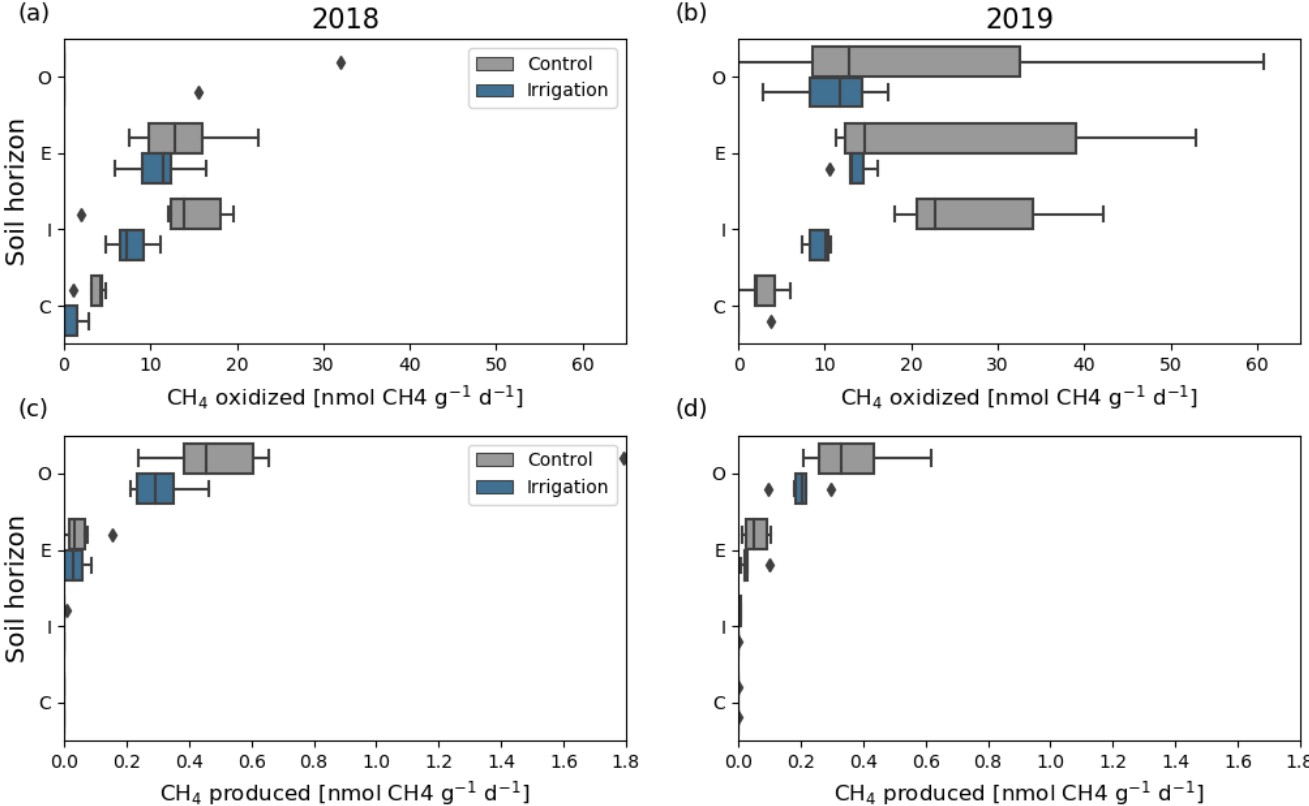





Figure 7. CH$_4$ oxidation (a, b) and production (c, d) potentials in different podzolic soil layers (organic layer, O; leaching layer, E; enrichment layer, I; bottom layer, C) in 2018 (a, c) and 2019 (b, d) (n=9). The boxes show the quartiles of the dataset and the vertical line inside the boxes is the median flux. Whiskers show the range of the data, except for the data that are determined to be outliers, which are shown with black diamonds.

**3.4 Factors controlling field CH$_4$ fluxes**

Correlations between field CH$_4$ flux and SM were nearly negligible ($\rho < 0.2$) in both S$_i$ and S$_c$ in both years (Fig. 8), with the exception of S$_i$ in 2019 with an $\rho$ value of –0.5 ($p < 0.001$). In both 2018 and 2019, correlation trends were weakly negative between CH$_4$ flux and SM, except for S$_c$ in 2018 with a weak positive correlation ($\rho = 0.17$, $p = 0.07$). In contrast, CH$_4$ flux and ST had generally stronger correlations in both S$_i$ and S$_c$, the latter having the highest $\rho$ values in both years (2018: $\rho = -0.57$; 2019: $\rho = -0.49$), only 2018, however, being statistically significant ($p < 0.001$). S$_i$ showed differing correlation trends between years, 2018 having relatively weak positive ($\rho = 0.4$, $p = 0.01$) and 2019 almost negligible negative correlations ($\rho = -0.19$, $p = 0.16$).

Several mixed-effect model runs were made to investigate the environmental drivers behind CH$_4$ fluxes. SM5cm and ST5cm were among the significant variables explaining CH$_4$ fluxes in all the model runs. The rest of the significant drivers varied depending on the soil layer. In the organic layer, the most significant model, in addition to SM$_{5cm}$ and ST$_{5cm}$, included oxidation potential and nitrogen content. The model explained 51 % of the variation in CH$_4$ fluxes (Table 1). The significant drivers were otherwise similar to O layer in E layer, except nitrogen content was replaced by carbon. However, the model had weaker explanative power ($r^2_{fix} = 0.44$) than the O layer model ($r^2_{fix} = 0.51$). It should be noted that carbon and nitrogen contents had strong cross-correlation, and using either of them in the model would have given almost the same result. The I layer model had the weakest model explaining CH$_4$ fluxes ($r^2_{fix} = 0.42$), and the significant drivers included only SM$_{5cm}$ and ST$_{5cm}$ and the carbon content in the I layer. Using the drivers' mean values over all soil layers also resulted in a relatively weak model ($r^2_{fix} = 0.44$), and it included only oxidation potential and SM$_{5cm}$ and ST$_{5cm}$. Finally, a model combining drivers from multiple depths was made, and it explained the CH$_4$ flux the best ($r^2_{fix} = 0.65$). In that model, CH$_4$ flux was most influenced by SM$_{5cm}$ and ST$_{5cm}$, oxidation potential in the organic layer, production potentials in the organic and E layer, and carbon content in the E layer (Table 1).



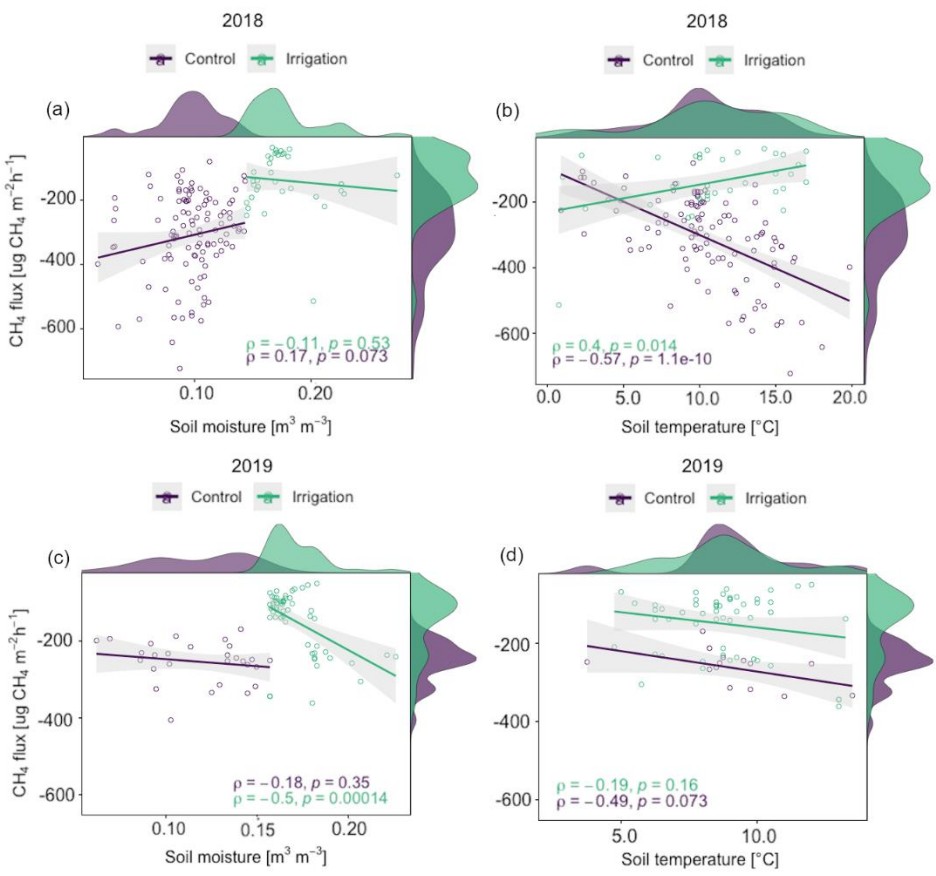

Figure 8. Correlations (Pearson's coefficient, ρ) between soil moisture and CH$_4$ flux (a, c) and soil temperature and CH$_4$ flux (b, d) with smoothed frequency histograms in 2018 and 2019. The emission case of 27 June 2019 was removed from the data in the correlation analyses for more clear presentation.

5    Table 1. Linear mixed-effect models fitted against CH$_4$ fluxes (F$_{CH4}$) and experimental factors. The fixed effects in the model were: SM – 5 cm soil moisture; ST – 5 cm soil temperature; OP$_{CH4, x}$ - CH$_4$ oxidation potential at soil layer x (O, E, I soil layers or the mean of all layers), PP$_{CH4, x}$ - CH$_4$ production potential at soil layer x, CC$_x$ - carbon content at soil layer x, and NC$_x$ - nitrogen content at soil layer x. The table shows the r$^2$ of the fixed effects ($r_{fix}^2$) and the whole model (fixed effects + random effects, $r_{mod}^2$), p-value of the model (p), AIC of the model and the degrees of freedom (df). The models in bold are the best-

10    fitted models.



| | Mixed effect model equations | $R^2_{fix}$ | $R^2_{mod}$ | p | AIC | df |
|---|---|---|---|---|---|---|
| **Mean of layers** | | | | | | |
| Model 1 | $F_{CH4} \sim SM+ST+OP_{CH4,mean} +PP_{CH4,mean}+Nmean$ | 0.38 | 0.78 | <0.001 | 1662.9 | 8 |
| Model 2 | $F_{CH4} \sim SM+ST+OP_{CH4,mean}+PP_{CH4,mean}$ | 0.414 | 0.761 | <0.001 | 1660.9 | 7 |
| **Model 3** | **$F_{CH4} \sim SM+ST+OP_{CH4,mean}$** | **0.435** | **0.744** | **<0.001** | **1658.9** | **6** |
| **Organic layer** | | | | | | |
| Model 1 | $F_{CH4} \sim SM+ST+OP_{CH4,O}+PP_{CH4,O}+NC_O$ | 0.496 | 0.771 | <0.001 | 1659.6 | 8 |
| **Model 2** | **$F_{CH4} \sim SM+ST+OP_{CH4,O}+NC_O$** | **0.513** | **0.758** | **<0.001** | **1658.1** | **7** |
| **E layer** | | | | | | |
| Model 1 | $F_{CH4} \sim SM+ST+OP_{CH4,E}+PP_{CH4\ E}+CC_E$ | 0.42 | 0.751 | <0.001 | 1660.3 | 8 |
| **Model 2** | **$F_{CH4} \sim SM+ST+OP_{CH4,E}+CC_E$** | **0.443** | **0.747** | **<0.001** | **1659.4** | **7** |
| **I layer** | | | | | | |
| Model 1 | $F_{CH4} \sim SM+ST+OP_{CH4,I}+PP_{CH4,I}+CC_I$ | 0.37 | 0.775 | <0.001 | 1662.8 | 8 |
| Model 2 | $F_{CH4} \sim SM+ST+OP_{CH4,I}+CC_I$ | 0.403 | 0.755 | <0.001 | 1660.8 | 7 |
| **Model 3** | **$F_{CH4} \sim SM+ST+CC_I$** | **0.415** | **0.743** | **<0.001** | **1659.3** | **6** |
| **Combination** | | | | | | |
| **Model 1** | **$F_{CH4} \sim SM+ST+OP_{CH4,O}+PP_{CH4,O}+PP_{CH4,E}+CC_E$** | **0.648** | **0.741** | **<0.001** | **1650.2** | **9** |

### 3.5 Microcosm experiment

Adding glucose to the sample and keeping the moisture level similar did not cause significant changes in the potential $CH_4$
uptake rate, but on average, $CH_4$ uptake rate was lower or the $CH_4$ emission was higher with added glucose on the same moisture level (Fig. 9). There was an exception to this case at high 25 °C temperature and moderate moisture (M1), where the added glucose samples had a higher $CH_4$ uptake rate, but as said above, these were not statistically significant differences.

Generally, increasing soil moisture with no added glucose decreased the mean potential $CH_4$ uptake rate, but even with the high SM (M2) group, the soil did not turn into a $CH_4$ source (Fig. 9). Also, the differences between the different moisture
groups were generally not statistically significant. Significant differences were only observed between the M2 and the control group in the first two weeks of measurements. The weekly mean $CH_4$ uptake rates also decreased further in time in all groups, except in the M2 group, where the changes in time were negligible.

In samples with added glucose, increasing SM significantly ($p < 0.05$) decreased potential $CH_4$ uptake rate in both M1 and M2 groups compared to the control group. On the other hand, M1 and M2 groups did not differ significantly, except in week two
at 25 °C temperature. In that case, relatively high $CH_4$ emission was measured in the M2 group with added glucose, but emission dropped rapidly already in the third week, although it remained a small $CH_4$ source (Fig. 8). At 15 °C temperature, there was no such $CH_4$ emission peak in the M2 group.



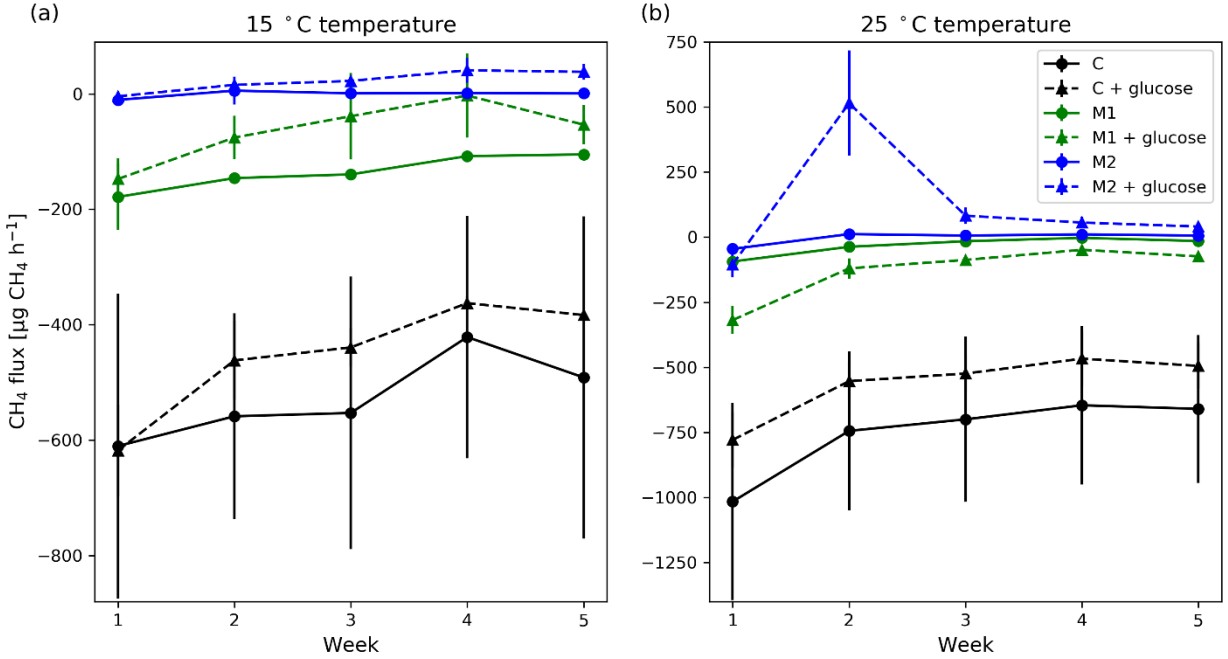

Figure 9. Weekly mean CH$_4$ flux measured at 15 °C (a) and 25 °C (b) without (solid lines) and with (dashed lines) added glucose on different moisture levels (black: control, green: low added moisture, blue: high added moisture). The error bars show the standard error of the mean (n=3).

## 4 Discussion

In this study, our initial aim was to mimic a wet growing season in a boreal upland forest with podzol soil in Northern Finland by irrigating the area regularly and studying the conditions needed to switch the forest floor to a CH$_4$ source. Earlier, we discovered that the soil of the same site turned into a CH$_4$ source in August after long-lasting rains during the growing season of 2011 (Lohila et al. 2016). Therefore, we assumed that we could reach the conditions needed to initiate the CH$_4$ production in the podzolic soil by at least tripling the long-term mean precipitation. However, the two study summers of 2018 and 2019 turned out to be the driest summers of the decade, with a long warm and dry period in June-July 2018 and generally dry summer in 2019. As a result, our control plot could be considered a drought experiment, while the irrigated plot followed the moisture and CH$_4$ flux patterns of a "normal" summer. In the Lohila et al. (2016) study, we also speculated that the reason for the CH$_4$ emission occurring in August and not in spring after the snowmelt could be that fresh carbon substrates consumed by soil microbes are needed to make the soil anoxic, i.e. the wet soil alone is not enough to make the CH$_4$ production to initiate. To confirm this hypothesis, we conducted a laboratory mesocosm experiment in which the temperature and moisture responses were studied, and glucose was added to some of the samples to mimic the root exudation providing fresh carbon substrates to the soil microbes.



We found that the field $CH_4$ uptake at the control plot was higher during the study years than a more typical summer of 2017. This comparison was possible since some of our study plots were established already a year before the experiment. On the other hand, the irrigated plot showed similar uptake rates during the previous summer, which was close to normal in terms of temperature and precipitation. The same pattern was observed for the soil moisture: the soil was as moist in the irrigated plot in 2018 and 2019 as it was without irrigation in 2017. One single occasion when clear $CH_4$ emission was detected took place in the irrigated plot at the end of June 2019, but the emission was only observed in one of the irrigated plots. The mean emission rate during that day from the irrigated plots was 1670 µg $CH_4$ $m^{-2}$ $h^{-1}$ (data not shown, the point removed from Fig. 5b). Although encouraging, the observation unfortunately did not provide means to systematically study the conditions needed to switch the soil into a $CH_4$ source, since the soil moisture or any other variable at the same measurement point did not differ from the other points.

The laboratory experiments for studying the possible differences in the $CH_4$ production and oxidation potentials indicated no significant differences between the control and irrigated soils. Initially, we hypothesised that the wet conditions prevailing for one or two summers could be seen in the response of microbial populations so that at the irrigated plot the oxidation would be smaller and the production higher either after both summers or at least after the second wet summer. Unfortunately, the dry summers turned the whole setup around so that we ended up examining the effect of dry growing seasons on the response of microbial populations. Hence, our results suggest that the period of one or two dry summers did not impact soil production or oxidation potentials, although we found differences in the actual $CH_4$ uptake between the irrigated and control plots. Therefore, it seems likely that the differences in observed field fluxes were due to the impact of soil moisture on the gas diffusion rate: the drier the soil, the higher the air-filled porosity and the quicker the diffusion of oxygen and $CH_4$ into the soils, and the higher the $CH_4$ uptake rates (Dörr et al., 1993; Van Den Pol-van Dasselaar et al., 1998; Striegl, 1993).

We also hypothesised that the oxidation potentials would be highest in the topsoil, which is closest to the main source of the substrate for oxidation, namely atmospheric $CH_4$ (Bradford et al., 2001), while the production potentials would be higher deeper in the soil, where the oxygen is more likely to be depleted periodically after wet conditions when diffusion rates are suppressed. The oxidation potentials indeed peaked in the topsoil, but interestingly, so did the production potentials, showing clearly the highest rates in the organic/humus layers.

Our findings are parallel with the previous ones from forest soils since the highest $CH_4$ oxidation has been detected both in the uppermost mineral soil below the organic layer (Saari et al., 1998) and, on the other hand in the organic layer (Wang and Ineson, 2003). Thus, the distribution of $CH_4$ consuming organisms in the upland soil horizon seems to vary somewhat depending on the year and prevailing physical and chemical conditions. High potential $CH_4$ productions in the surface layers in 2018 and 2019 are most likely linked to higher soil organic matter and moisture content of soils, which is also supported by a strong positive correlation with the soil organic matter and moisture content. Potential $CH_4$ oxidation did not show a strong correlation with these. Similar results obtained from upland soils and especially from forest soils are hard to find. However,





high organic C content has simulated CH$_4$ production under hypoxia in agricultural soil (Brzezińska et al., 2012), and water content was observed as a major influencing factor regarding CH$_4$ production potential in subalpine upland soil (Praeg et al., 2014). Thus, over two times higher moisture content and about ten times higher organic matter content in the organic layer compared to mineral layers below most likely explain partly the higher CH$_4$ production potentials observed in this study.

The mesocosm experiment provided interesting insights into the CH$_4$ dynamics of the podzolic soil. First of all, this experiment confirmed the result of field fluxes by showing that the CH$_4$ uptake decreased along with higher soil moisture. Also, CH$_4$ uptake was totally ceased in the high soil moisture treatment (M2) due to suppression of diffusion rates in waterlogged conditions. The higher temperature increased the net uptake, most likely by increasing the oxidation, but this was true only for the mesocosms with "field conditions" (no water added). In other words, CH$_4$ uptake was higher in warmer soils but smaller
in wetter soils (as expected), so it seemed that increasing either the temperature or the soil moisture, or both, affect the CH$_4$ oxidation straightforwardly but is not able to induce CH$_4$ production in the soil. However, only if the soils were made wet enough and glucose was added, significant CH$_4$ production was initiated, which was further increased by higher temperatures. Thus, the results obtained here supported our hypothesis that both excess moisture and easily decomposable carbon are needed to initiate CH$_4$ production in podzolic soil. Indeed, the root exudate analogues containing simple sugars accelerated CH$_4$
production in tropical peat soil (Girkin et al., 2018). However, in a study conducted in Japanese upland soil, added glucose was rapidly decomposed within seven days of the incubation, and part of the glucose-derived C flow ended up to methanogens even under unflooded conditions (Watanabe et al., 2011). Even though it is largely known that methanogens can survive and tolerate dry and oxic conditions for some periods, they become active only when the conditions turn favourable for CH$_4$ production (i.e., wet and anoxic). Since methanogenic archaea cannot use glucose directly as C source, methanogens probably
utilised acetate or CO$_2$ produced by the glucose-decomposing bacteria. Thus, the obtained results from the microcosm experiment may reflect the situation that in wet conditions, glucose has increased the activity of microbial communities that supply methanogenic substrates (hydrogen-producing bacteria or acetyl-producing bacteria), promoting the activity of methanogens production as was detected in a forested wetland (Koh et al., 2009). Simultaneously decreased activity of CH$_4$ oxidisers may have been followed by the competition of other aerobic microorganisms, which have metabolised glucose
rapidly, creating more anaerobic conditions favouring CH$_4$ production. However, the comparison of the obtained results with earlier findings is rather obscure since similar experiments conducted in boreal forest soil do not exist. Thus our results are one of the first attempts to understand the complex conditions which initiate CH$_4$ production. In addition, our study is unique since we are presenting both CH$_4$ fluxes and laboratory CH$_4$ potentials from the soil taken from the same field points.

## 5 Conclusions

Based on our field and laboratory experiments, the main conclusion is that CH$_4$ production from boreal upland forest soil cannot occur solely by prolonged wet conditions, but there also has to be enough fresh carbon in the soil. Therefore, we expect

the possible $CH_4$ production episodes to occur in late summer and autumn rather than in spring, even though the soil can be very wet after snowmelt. These findings can be applied in $CH_4$ process models to improve estimations of regional and global upland forest $CH_4$ balances.

We did not observe any changes in $CH_4$ production and oxidation potentials due to irrigation over two summers, meaning microbial communities were not very sensitive to environmental variables. This suggests that the measured field fluxes are rather controlled by the physical soil conditions by limiting gas diffusion rates and not by the changes due to microbial function. One conclusion from our results is that $CH_4$ production and oxidation are controlled by different driving variables and processes: the oxidation is boosted when the conditions for higher gaseous diffusion are optimal (dry soil), while the production is boosted only if anoxic conditions are created (wet soil reducing diffusion + microbial activity consuming oxygen) and there are fresh organic substrates available for $CH_4$ production. In our field experiment, $CH_4$ production episodes were not detected (without one exception), and the changes in net field $CH_4$ flux were solely caused by the changes in $CH_4$ oxidation. The net $CH_4$ flux (here total oxidation) was primarily controlled by soil temperature and soil moisture. Increasing soil temperature enhanced oxidation and gas diffusion while increasing soil moisture limited oxidation by making conditions for methanotrophs unfavourable and diminished diffusion. We also found that upland forest soils have the potential to produce $CH_4$, but contrary to wetlands, the potential is highest near the soil surface and decreases rapidly as a function of soil depth. This could happen because the conditions for methanogenic archaea are more favourable in the topsoil layer due to the higher amount of organic matter.

Our study confirms that soil moisture is a critical variable in explaining the soil $CH_4$ uptake rate and suggests that the diffusion rate of both $CH_4$ and oxygen into the soil is the primary constraint of oxidation. For the onset of $CH_4$ production in podzolic soil, not only high soil moisture but also the addition of sugar, mimicking root exudates from trees, was needed. Glucose impacts $CH_4$ production mainly by boosting the consumption of oxygen in the soil and providing substrates for $CH_4$ production. We also found that the highest potential production and oxidation rates were found in the same topsoil layers, suggesting that the surface soil plays the main role in the soil-atmosphere exchange of $CH_4$ in boreal upland forest

**Data availability**

The flux, meteorological and soil data are available at Zenodo (https://doi.org/10.5281/zenodo.5153347, Korkiakoski et al., 2021)

**Author contribution**

AL, TP and KP designed the study. MK, AL, TP and TM constructed the experimental site. KP took the soil samples and calculated methane production and oxidation potentials and did the laboratory work for the microcosm experiment. TM made





the flux measurements and took part in the data analysis. MK calculated the fluxes for the field and microcosm experiments and did the statistical analysis. MK prepared the manuscript with contributions from all co-authors.

**Competing interests**

The authors declare that they have no conflict of interest.

**Acknowledgements**

We thank Academy of Finland for funding this research (Grant no. 308511). Valtteri Hyöky, Päivi Pietikäinen, Stephanie Gerin and Petri Salovaara are acknowledged for assisting in the fieldwork. We also want to thank Dr. Bastian Steinhoff-Knopp for letting us use his aerial photo of the experimental site.

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
