# Peer review of "Excess soil moisture and fresh carbon input are prerequisites for methane production in podzolic soil"

_Biogeosciences, 2021_

## Author Comment (AC1)

**We want to thank the referee for the positive and encouraging comments. See our numbered responses (bolded) to the comments below.**

**On behalf of all authors,**

**Mika Korkiakoski**

**Referee #1**

Recent reports on greenhouse gases budget reveal that CH4 emissions have been increasing over the past decade, raising concerns for the role of CH4 on global warming. Warming may also contribute to the increase in CH4 emissions. As the authors mention, high latitude regions contribute substantially to CH4 emissions due to the large areal extent of wetlands. Although forest soils partially offset such emissions by oxidizing some of the atmospheric CH4, it is very important to understand how such a balance will be altered by warming. In this regard, this manuscript is timely and addresses an important question, which --in my opinion-- is relevant to the readership of this journal and beyond.

The paper is well written and clear. I only have a couple of concerns that I believe the authors should address for the sake of clarity.

It is not clear the amount of water that is being delivered through irrigation and how this amount was decided. Because the research question has to do with soil water content, it seems logical to me that the irrigation water should be decided based on the soil water content. Furthermore, to really trigger methane production (as well as other anaerobic processes), it is well known that the soil water content should rise above the soil water at field capacity. For methanogenesis, we also know that with soils at (or almost) saturation, it still takes 1 to 2 weeks (sometimes more) before methane starts being produced. There is a lot of literature from rice cultivation on this, for example. Thus, with irrigation active only once a week it is not surprising that methane production was low, or perhaps even absent, from the plots.

**1. Concluding from this and the other referee's comment (see our answer #10), we see that the description of the irrigation can be confusing (page 4, lines 22-25). To clarify, we rewrote those lines as follows and hope it is clearer now: "The amount of rainfall added with irrigation was 11 mm on two days a week during 28 May–1 June 2018, after which the amount was increased to 11 mm on three days a week during 7–18 June 2018 and eventually to 21 mm on five days a week during 20 June 2018–7 September 2018. In 2019, the plot was irrigated with 11 mm three times a week throughout the summer."**

**Generally, in 2018, we irrigated the plot five times a week (every weekday) with 2000 liters of water. In the first half of June, we started with a lesser amount but noticed that it would not be enough as the summer turned out to be exceptionally dry. Initially, the amount of irrigation was decided to double the amount of summertime rainfall. However, practically, we had to vary the irrigated amount based on soil water content and what was practical. We did not know the exact soil field capacity value, so we could not base our irrigation amount on that. Irrigating five times a week with 2000 liters was pretty much the maximum amount we could do because**

**we had one 1000 liter tank that had to be towed for almost 6 km on top of a hill on small dirt roads twice each workday. Daily irrigation with 2000 liters took 4 hours to do and it had to be timed at late afternoon/early evening to minimize the evaporation of the irrigated water. In 2019, we, unfortunately, had to decrease the amount of irrigation to three times a week with 1000 liters at a time due to practical reasons.**

Also, rainfall (which this irrigation aimed to mimic) is intermittent but not at regular intervals. This is very important for biogeochemical processes. The fact that sometimes two rain events happen very close with each other may cause soil water content to reach very high values, necessary for anaerobic processes. An equal amount of rainfall delivered over regular interval may not lead to very high water contents. This may also explain why in a wet year you can observe CH4 production, whereas an equal amount of water delivered more regularly through irrigation may not lead to CH4 production.

**2. See also our answer #1. The irrigation was actually done at regular intervals and not intermittently.**

**The problem was that especially 2018 was so dry that the soil water content had usually returned to the pre-irrigated level before the next irrigation on the following day, which happened likely because of evapotranspiration of a significant amount of irrigated water even though we did the irrigation on late afternoon/early evening. The soil water content usually returned to the pre-irrigated level even with 2000 liters of daily irrigation, which corresponded to 21 mm of precipitation (page 4 lines 29-30).**

**We were aware that, to initiate net CH4 emissions, we will probably need to reach both a high soil moisture level for a longer period and, in addition, have enough fresh carbon substrates for the methanogenic archaea. We expected that by conducting regular irrigation, the soil water content would gradually increase and by the end of the summer, we would have a higher base level (the lowest points of the highly variable soil moisture values) of soil water content. Then, with the low diffusion of oxygen into wet soil + the irrigation, we could create anoxic conditions for longer periods and promote CH4 emissions. We expected CH4 emissions to occur somewhere in August when fresh carbon substrates were available and soil moisture content would be high, resulting from our regular irrigation mimicking a wet summer. This was discussed on page 20 lines 6-18.**

Lastly, if the authors wanted to find the threshold in soil water content (or rainfall) leading to soils becoming a CH4 source, why not increasing irrigation (in frequency and amount)? This is also connected to my first point. It should be the soil water content to dictate the amount of water needed for soils to become anaerobic. I find this an interesting question and it is very unfortunate that the authors were not "successful" in finding this rainfall threshold. It would have been a very useful result, but I do not follow entirely why the authors did not explore higher water contents and irrigation water amounts.

**3. See also answers #1 and #2 above. The referee is right that by increasing irrigation frequency and its amount, the experiment may have led to finding a threshold for methane production but, unfortunately, due to the location of the study site on a remote upland forest, many kilometres to the closest water tap, we had limited resources for transporting the required amount of water into the site. In future replications of this experiment, it is definitely recommended that the experiment is set in a location where the irrigation system can**

distribute higher amounts of water and at a higher frequency than what was practically possible in this experiment.

**We modified lines 12-13 on page 20 as follows: "... summer in 2019. Unfortunately, due to the remote location of the experimental site and a distance of several kilometres to the closest water tap, we were not able to counter the effect of the droughts. As a result, our control plot could be considered a drought experiment, while the irrigated plot followed the moisture and CH4 flux patterns of a "normal" summer. Therefore, it is definitely recommended that in future similar studies, the experiment is set in a location where the irrigation system is able to distribute higher amounts of water and with higher frequency than what was practically possible in this study. In the Lohila et al. (2016) study…"**

I believe the authors should discuss these points in depth for a stronger and more sound paper. Hope my comments are useful.

**4. We want to thank the referee for the encouraging and helpful comments. They definitely helped to improve the paper!**

Minor points:

The need of labile C (e.g., sugars) is also needed from an energetic perspective, as anaerobic processes in general are not very favorable thermodynamically.

**5. The referee is absolutely right about that, but we think that it is not that relevant in our manuscript and have decided not to discuss it further.**

There are also recent methane budgets to consider. See for example: https://essd.copernicus.org/articles/12/1561/2020/

**6. To indicate the contribution of northern wetlands and mineral soils to the global CH4 budget in more detail, we modified the end of the paragraph on page 3 lines 1-2 as follows:"... those from northern peatlands, which contribute ~2 % to global CH4 emissions (Lohila et al., 2016; Saunois et al., 2020). Even though the global CH4 uptake by mineral soils is only ~5 % of global CH4 sink (625 Tg CH4 yr-1; Saunois et al., 2020), it has already been suggested…"**

---

## Author Comment (AC2)

We want to thank the referee for the positive and encouraging comments. See our numbered responses (bolded) to the comments below. Note that the numbering continues from our answers to Referee #1.

On behalf of all authors,

Mika Korkiakoski

**Referee #2**

Korkeakoski et al. investigated the dynamics of methane production and oxidation at a field manipulation experiment subjected to an irrigation treatment. However, the field manipulation experiment occurred against the background of a strong drought, so the results were perhaps not as clear as they had originally hoped. To investigate the processes contributing to methane flux, the authors also undertook two separate laboratory experiments: soil incubations to quantify rates of methane production and oxidation and mesocosm experiments with a glucose addition to simulate the effects of plant root exudates on methane production and oxidation.

Overall, I found this to be a very nice, convincing manuscript. It was nicely written, relatively easy to follow, and the interpretation of the results was on-point. My greatest concern was with two of the statistical analyses: the pearsons correlation and the mixed effects model for environmental drivers. With the statistics, you want to be testing something that is biologically meaningful; how biologically meaningful is a correlation between oxidation and copper concentrations? I think this analysis should be constrained to predictors that are biologically meaningful. Similarly, the mixed effects model seemed to not capture the most important drivers of methane oxidation rates: the diffusion limitiation of methane into the soil. This is discussed directly later, so I am not so concerned about this.

**7. Our statistical analysis can be divided into two parts: using Pearson correlations (production and oxidation potentials, and field fluxes) and mixed-effects modeling (field fluxes). In the case of the field fluxes, the correlations were checked only against soil moisture (which limits gas diffusion) and soil temperature (controls the biological activity of microbes), because those are well-known to control soil CH4 exchange, and those were the only ones that were always measured alongside the fluxes while other parameters were measured only once a year. In the mixed-effects model, we included the most significant predictors that could explain variations in CH4 fluxes. We used: soil moisture, soil temperature, oxidation and production potentials and soil C and N contents, which we thought were possibly the most significant predictors to explain the measured CH4 fluxes. We did not use oxygen or any nutrient concentrations in the mixed-effects model analysis.**

**However, in the correlation analysis concerning the production and oxidation potentials, we tested potentials against a wide variety of predictors such as different nutrients. We understand the referee's argument that this is a problem because many of them are not necessarily biologically meaningful.**

**We have decided to remove following text from the results (p.16 lines 8-9): "and with several nutrient elements such as with sulphur ($\rho = 0.93$), lead ($\rho = 0.92$) and potassium concentrations ($\rho = 0.81$)".**

**We also removed the sentence (p. 16 line 11-12): "Potential CH4 oxidation rates showed only a negative correlation with measured copper concentrations ($\rho = -0.61$)."**

One note on the presentation. I struggled a bit with the figures because many of them were quite busy and didn't transfer well to black and white (I always print things to review them). Looking at them again online, even having them in color doesn't help so much. Color palettes from R-color brewer might help? Beyond my complaints about the figures, I really only have minor comments and a few technical corrections as outlined below and really enjoyed reading the manuscript.

**8. The referee is correct that many of the figures could be difficult to interpret if printed in black and white. We tried to modify the figures to take these facts into account. See our answers #15-20 below.**

Specific comments.

p.2 line 19: Technically the boreal samples in Blazewicz were from a rich fen.

**9. The referee is correct. We removed this line and reference.**

p.4 lines 20-25: change to "per week". I didn't fine these numbers so helpful, the ones on lines 29-30 seemed much more relevant. Rain input in mm is much easier to interpret, I think this would be better to present when discussing the treatment and the specific liters could be given in parentheses. Also nice would be the total amount of water added.

**10. We agree that indicating the amount of irrigated water is easier to understand in mm than in liters and we decided to change it as suggested by the referee. The other referee found the same part of the text confusing, and the exact change to the text is explained in answer #1 (see our responses to Referee #1).**

p.5 Could add that 2017 was monitored as a baseline prior to beginning experimental treatments to check for initial differences between the plots.

**11. We added a sentence: "The flux measurements in 2017 were made to check possible differences between the experimental plots before starting the irrigation experiment in 2018."**

p.8 line 1. I was confused about what microbial C has to do with any of this, why this as a reference amount? Maybe just specify the amount of glucose added?

**12. The purpose was to explain that we will offer excess carbon in a form of glucose for microbial use in relation to the carbon that microbes are storing in their biomass, nothing else. The amount of glucose was 14 ml 1 M glucose solution for each jar. This is now specified in Section 2.5 of the manuscript.**

Results

Section 3.1: What I missed here was directly addressing if there were pre-treatment differences in the environmental conditions between the experimental plots. Figure 6 seems to indicate that this is unlikely as the behavior is similar, but would be nice to address directly (if possible).

**13. In section 2.8, we described general weather conditions (air temperature, precipitation and snow cover) between the years (2017-2019), and we think that the differences between the years are clearly mentioned. This section can be considered as background data from a weather station (located at the same hill as the experimental site), which is why it is located in mat&met and not in the results section.**

**In section 3.1, we reported measurements of two different soil moisture profiles located just outside the experimental area (the location of these profiles will be added to Fig. 1) and 5 cm soil moistures and temperatures recorded next to the flux measurement points. Unlike the data described in section 2.8., these were part of our experimental setup and they are therefore part of the results. We think that soil moistures between pre-treatment and post-treatment conditions are already described well enough in the first paragraph of section 3.1. However, we could not compare soil temperature of the pre-treatment year against the post-treatment years because we did not have proper soil temperature measurements before the experiment (in 2017) to do that.**

p.14 line 15: I found this really confusing: position on the plot is higher, uptake rate is smaller, is flux higher?

**14. Yes, we can see how it is confusing. We changed the sentence indicated by the referee to: "The median measured CH4 uptake (Fig. 6; Table S5) across all the measurement points at the Si (150 ug CH4 m-2 h-1) was 48 % lower than at the Sc (290 ug CH4 m-2 h-1) in 2018 and 35 % lower in 2019 (Si: 170 ug CH4 m-2 h-1, Sc: 260 CH4 m-2 h-1)."**

Figure 3. Hard to tell the lines apart and figure out what the figure is showing.

**15. We changed the formatting of the lines and hopefully they are more distinguishable now. See the modified figure below.**

[Figure]

Figure 4. could improve with adding indication of precipitation. There are a lot of lines, again a bit visually challenging.

**16. The point of this figure was to show how the soil moisture behaves during and after irrigation compared to the soil moisture at the control. We did not want to draw ex. the mean and deviation of soil moisture at the irrigation and control plots, because it would not properly show the effect of irrigation due to differences in the moisture level between the measurement points. The spikes in the soil moisture lines would get dampened if means were calculated. Also, we decided to plot all the soil moisture data, so we did not need to subjectively select one or two lines to draw from the irrigated and control plot. The individual moisture lines are not important in this figure, which is why it does not matter if they cannot be clearly distinguished. We added an extra figure citation on page 11 line 23-24 to highlight why the figure is included in the manuscript: "However, the SM$_{5cm}$ decreased fast and usually returned to the pre-irrigation level before the next irrigation 24 hours later (Fig. 4)."**

**It is not explicitly stated, but in the figure, when the soil moisture increases at the control plot, it indicates precipitation.**

Figure 5. The combination of flux and soil temperature here is really difficult to see anything and even to distinguish between the lines. This one is important but it doesn't work.

**17. We moved the flux lines below the temperature lines to make them more distinguishable. See the modified figure below.**

[Figure]

Figure 6. panels a and b could easily have same axis, so why don't they? Also helpful would be an indication in panel a that there was the pre- and post-treatment period, and an indication of net uptake or net realease.

**18. The referee is right about the axis and it is now fixed. We added an indication of pre-treatment and pre-treatment periods to the panels. We also added a clarification about net uptake and emission into the figure caption. We also modified the colors of the boxes so that they match with Fig. 7 (See answer #19). See the modified figure below.**

[Figure]

Figure 7: panel a: where are the boxes for O treatments? panel c: is that a point way over on the right side nearly under the figure legend? It should really be a bit better shown on the plot. I would also suggest using gray and open boxes for this figure to improve readability.

**19. The box for O layer in panel a is invisible because the oxidation rate was very small except in those two cases (diamonds), which were considered as outliers. This was mentioned in the manuscript on page 15 line 10.**

**We have fixed the legend positioning in panel c and changed the coloring of the boxes in the figure as suggested. See the modified figure below.**

[Figure]

Figure 9. Note different y-axis scales again, CH4 flux seems to be missing part of the unit. Colors difficult to distinguish. Again, this indication of net emission or net uptake could be helpful.

**20. Yaxes are now fixed and we also changed the coloring and added a note about the net emissions and net uptake into the figure caption.**

**The referee is right that the ylabel was missing part of the unit. We also transformed the units so that they are now the same as those of the field fluxes. See the modified figure below.**

[Figure]

Discussion: really nice.

p.21 lines 20-25 could use a figure reference plus it could help to tie some of this to the newer soil literature that discusses the spatial distributions of microbes and substrate as a key control on soil processes (partly summarized in the Schmidt et al. 2011 Science paper).

**21. We are aware that spatial distribution of microbes and substrate availability are key controls on soil processes and that there will be references on that matter. However, our purpose was to find references and discuss specifically on distribution of methane-cycling microbes only (oxidizers and producers), and their vertical distribution in mineral soil is not too much studied.**

p.21 28-32: this was the only part that I didn't think was so well supported, both the part about the distribution of organisms (what if it's active vs. dormant?) and I don't see from Figure 8 a strong correlation between flux and soil moisture.

**22. This discussion part about the distribution of organisms refers to the results from potential oxidation/production rate values measured in the lab and is shown in Figure 7.**

p.22 line 20: don't forget about the other fermenters! E.g. Tveit et al. 2013

**23. Sure, there might be other important players/fermenters, but we are dealing with mineral and not organic soils, so we prefer to speculate how added glucose might end up to facilitate methanogens via glucose-users first, which are creating suitable conditions for methanogens to act in mineral soils like our forest soil.**

p.23 line 11: „with one exception"; line 13: "while" should be "and"
**24. Corrected.**

---

## Author Response (AR2)

Dear Editor,

See below our changes to the text based on your comments. Some of the changes may not show properly in this file and we recommend you to check also the manuscript pdf file.

On behalf of all co-authors,

Mika Korkiakoski

[revised manuscript text omitted]